# Air Pollution Management: A Multivariate Analysis of Citizens’ Perspectives and Their Willingness to Use Greener Forms of Transportation

**DOI:** 10.3390/ijerph192114613

**Published:** 2022-11-07

**Authors:** Silvia Puiu, Mihaela Tinca Udriștioiu, Liliana Velea

**Affiliations:** 1Department of Management, Marketing and Business Administration, Faculty of Economics and Business Administration, University of Craiova, 200585 Craiova, Romania; 2Department of Physics, Faculty of Sciences, University of Craiova, 200585 Craiova, Romania; 3Department of Humanities, University Ca’Foscari, 30123 Venice, Italy; 4National Meteorological Administration, Sos. Bucuresti-Ploiesti 97, Sect 1, 013686 București, Romania

**Keywords:** air pollution management, air pollution, air quality, public management, multivariate analysis, green investments, financial incentives, electrical cars, public transportation, urban gardens

## Abstract

The present research aims to understand how air pollution can be managed by public authorities, both central and local, starting from citizens’ perspectives on the issue. Air quality is a real problem, affecting people at multiple levels. Thus, we introduced the following variables to better understand the problem and to be able to formulate theoretical and practical implications for public management: the involvement of authorities in reducing air pollution; the involvement of citizens in reducing air pollution; financial incentives for citizens and companies for adopting behaviors that reduce air pollution; green investments in the city; the impact of air pollution on the community; and the need for independent bodies to monitor air pollution. The research methodology used is partial least squares structural equation modelling (PLS-SEM) and the required data were gathered from issuing a survey to citizens from the most important cities in Romania where pollution poses important challenges for the community and for the authorities. The results are useful to public managers in local and central institutions for creating better strategies meant to reduce air pollution, increase air quality, and improve the quality of the citizens’ lives.

## 1. Introduction

In the context of climate change and the increased importance of the sustainable development goals in the Agenda 2030, we focused our research on the eleventh (“Sustainable cities and communities”) and the thirteenth (“Climate Action”) goals [1]. The responsibility for the quality of our lives belongs to both authorities and citizens. Many studies present the solutions available for public managers, and the strategies and the policies they might use to reach these goals in their communities. In the present paper, we take a closer look at what citizens want and expect from local and central authorities.

According to the World Health Organization [2], “air pollution is contamination of the indoor or outdoor environment by any chemical, physical or biological agent that modifies the natural characteristics of the atmosphere”. The statistics from the World Health Organization [3] are worrying: more than 99% of people live in regions where the air quality is low, and more than four million people died in 2016 because of outdoor air pollution.

One of the most important polluters is represented by cars, which are responsible for “a fifth of EU emissions” [4]. Thus, our research analyses the willingness of citizens to change the way they travel and use greener solutions (public transportation or switching to electrical vehicles).

Public authorities can act on multiple levels: sanctioning polluters and incentivizing those who invest in greener solutions, such as using electrical vehicles, developing green buildings, expanding the green locations in the city, and expanding the number of charging stations for electrical cars. The solutions are numerous but a good strategy for managing air pollution should take into account the reluctance to change exhibited by citizens, which is sometimes motivated by limited financial resources. This is the reason we suggest that all strategies should have the citizens’ perspective as a foundation. A mix of financial incentives, changes in legislation, partnerships with the private sector, and also a proactive attitude from public authorities will encourage citizens and companies to change their behavior and contribute to the reduction of air pollution.

## 2. Literature Review

The novelty of our study consists in the role played by the citizens’ perspective for establishing strategies that will be easier to implement, and for which the reluctance to change will be reduced. Thus, we focused our research on the following variables, all seen through the citizens’ perspective: the involvement of authorities in reducing air pollution (monitoring, legislation, and controlling and implementing punitive measures); the involvement of citizens in reducing air pollution; financial incentives for citizens and companies for adopting behaviors that reduce air pollution; green investments in the city (green locations, urban gardens, green buildings, electrical vehicles, a network of charging stations for electrical cars, and green technologies); the impact of air pollution on the community (quality of life, health burdens, and touristic attractiveness); and the need for independent bodies to monitor air pollution.

Depending on the focus, there are two terms used frequently: air quality management focusing on the goal, and air pollution management focusing on the problem for which public authorities should find the best solutions to be implemented and accepted by the community. According to Environmental Protection [5], “air quality management refers to all the activities a regulatory authority undertakes to help protect human health and the environment from the harmful effects of air pollution”.

### 2.1. The Involvement of Authorities in Reducing Air Pollution (Passive Involvement)

Public authorities should monitor air pollution, adopt the needed legislation for reducing pollution, develop control activities, and sanction polluters. This variable reflects passive involvement; meanwhile, financial incentives might bring a more proactive attitude from citizens and organizations.

Huang et al. [6] analyze the role of the government in tackling the pollution problem and show that increased attention to environmental issues (consisting of laws, control mechanisms, establishing standards, and sanctioning polluters by imposing high taxes) leads to a decrease in pollution levels. Zou et al. [7] state that air pollution can be managed, but only through a concerted effort from both the government and citizens. The authors emphasize the importance of both perspectives and actions at a macroeconomic level and a microeconomic level.

Gwilliam et al. [8] focus on one of the most important contributors to air pollution, which is represented by city transportation. The authors mention multiple actions that can be undertaken by the government to reduce outdoor pollution: fiscal measures, changes in legislation, and educational campaigns to promote greener behaviors and examples of good practices that help with air quality.

Mahmood [9] (p. 139) presents the gravity of air pollution and the negative impact of air pollution on the health of people in Bangladesh; this study highlights how, despite the government’s efforts, the situation is not improving significantly. The author emphasizes that public managers should adopt stricter regulations to reduce air pollution and raise air quality, and also “translate its National Environmental Policy… into action”.

Jafari et al. [10] (p. 1911) classify public policies regarding air pollution management in three categories: “incentive”, “supportive”, and “punitive policies”. We assume that financial incentives and the support offered to the citizens and companies have a more powerful impact than punitive measures on changing citizens’ behavior. Thus, we decided to separate these forms of involvement from those concerning authorities to better understand the differences in the impact they have.

### 2.2. Financial Incentives for Citizens and Companies for Adopting Behaviours That Reduce Air Pollution (Active Involvement)

This type of support can consist in exempting citizens and companies of paying some taxes or offering a reduction in taxes, or in offering financial aid to facilitate changes in behavior by increasing public expenses. Wang and Wheeler [11] (p. 174) state that “pollution control through financial incentives has a much greater impact”, a statement with which we agree. Jacobs [12] (p. 113) also shows that this form of support was “unknown” at the end of the 1980s in the United Kingdom, because most forms of involvement from authorities referred to regulations, control, and sanctions for illegal behaviors.

The relationship between financial incentives and the changes in the behavior of citizens in using more eco-friendly means of transportation was also studied by Hutton and Markley [13]. They concluded that the pilot study was a success because most employees in their sample gave up using their personal cars for going to work and switched to public transport or ride sharing. Financial support for using electrical vehicles, for example, is consistent in many countries, but their success also depends on the purchasing power of individuals, their educational level, and the adequacy of the infrastructure for charging these cars [14]. Other authors [15] highlight the fact that these financial incentives should be high enough to make individuals and companies change their behaviors towards more sustainable forms of transportation.

### 2.3. The Involvement of Citizens in Reducing Air Pollution

The literature review uses various terms related to citizens’ involvement in reducing air pollution, such as citizen engagement [16,17], citizen participation [18], public engagement [19], or public participation [20]. Van Brussel and Huyse [16] conducted a study on the involvement of citizens in the development of public policies meant to reduce air pollution in Antwerp, and concluded that the people in the project became more responsible, changed their behaviours regarding the transportation they used, and participated actively in debates on these issues. Therefore, taking measures to influence citizen involvement is an important step that public managers should take.

Li et al. [20] appreciate that the involvement of citizens in reducing air pollution is not as developed in China as it is in countries in the West. Thus, the authors acknowledge the need to combine efforts at all levels (public authorities, private initiatives, and measures taken by individuals) to improve air quality. The authors also highlight the fact that this objective is meant to lead to the achievement of sustainable development goals.

### 2.4. Green Investments for Reducing Air Pollution

This concept refers to both public and private investments in green locations, green technologies, green means of transportation, green buildings, and urban gardens. Some authors use the term “green infrastructure” [21] to point out the important inequalities in the world in terms of the quality of the air we breathe.

Eyraud et al. [22] (p. 5) define green investments as “the investments necessary to reduce greenhouse gas and air pollutant emissions”. Green investments are influenced by “economic growth”, fiscal policies, and “fuel prices” [23]. Liao and (Roc) Shi [24] (p. 555) mention that one factor that leads to higher green investments is “public appeal”, which influences public authorities in establishing environmental strategies.

### 2.5. The Impact of Air Pollution on the Community

Pollution negatively impacts the quality of everyone’s lives, posing important risks to our health, increasing the costs in the health system [25], and even affecting incomes from tourism. As Sierra-Vargas and Teran [26] (p. 1031) put it, “air pollution is becoming a major health problem that affects millions of people worldwide”. Manisalidis et al. [27] argue that air pollution has a negative impact on health, leading to numerous illnesses (respiratory disease, heart problems, and even cancers) and suggest that the solution should consist in awareness campaigns intended to help people to understand that pollution is a real danger.

According to Afroz et al. [28], road transport is one of the most significant contributors to air pollution (almost 75% of the total emissions in Malaysia) and it leads to numerous problems for people’s health. Brauer et al. [29] emphasize the risks for the generations that are born these days, with pollution being a factor that affects the normal development of babies. As such, environmental policies should consider the risks for both present and future generations.

The impact on physical and mental health is mentioned in many other studies [30,31]. However, the negative impact of air pollution on the tourism sector is also an important reality, which affects many regions. Zhang et al. [32] conducted research on some of the most important cities in China and noticed a direct and negative relationship between air pollution and the number of tourists. Sajjad et al. [33] (p. 12403) even describe the problem of pollution as a “nightmare for tourism”. Zhou et al. [34] show that tourists also air quality take into account when they decide where to spend their vacation. Similar results are also presented in other works [35,36,37]. All of these findings should motivate public authorities, private organizations, and individuals in a community to act on reducing air pollution.

### 2.6. The Need for Independent Bodies to Monitor Air Pollution

Taking into account the impact of pollution on people, it is important to have a network of sensors to monitor pollutants and air quality. These values might be used by tourists, for example, when they decide on a touristic destination. The need for independent bodies to monitor pollution might arise from trust issues regarding public authorities. Both public and private networks of sensors are beneficial, as they give people the ability to compare data and make more informed decisions.

Alvear et al. [38] presents solutions such as “unmanned aerial vehicles” for monitoring air pollution, even in regions where this might be difficult. Meanwhile, Sivaraman et al. [39] (p. 56) emphasize some technical problems that reduce the reliability of the data from sensors, because “data on pollution concentrations is spatially too sparse”. The need for these sensors and the recent technological improvements are expressed in many studies [40,41,42,43]. Kaivonen and Ngai [44] emphasize the importance of having sensors that can offer data in real time and present the possibility of using them on public vehicles in motion.

The novelty of our research consists of our focus on citizens’ perspectives regarding air pollution management. We believe that this perspective can help authorities in raising the level of compliance when establishing public policies meant to protect the environment. Even if the proposed perspective is not analyzed by other researchers in its entirety, many studies present the success of environmental policies when authorities take citizens into account. Various countries might require different actions: Mahmood [9] shows that more strict regulations are needed in Bangladesh to make citizens comply; Wang and Wheeler [11] studied the connection between fiscal incentives and the changes in citizens’ behaviour and the effect on air quality in China, identifying a direct and positive relationship; Van Brussel and Huyse [16] also studied the role played by citizens in reducing air pollution in Antwerp. All of these examples show the important role played by the citizens, and demonstrate that authorities should construct their policies and strategies starting from the population.

## 3. Research Methodology and Hypotheses

The research methodology we used for this study is partial least squares structural equation modelling (PLS-SEM), with the help of the SmartPLS software, version 4 [45]. For the present research, we developed the following hypotheses:

**Hypothesis 1 (H1)**.
*The involvement of authorities in reducing air pollution positively influences the involvement of citizens in reducing air pollution.*


**Hypothesis 2 (H2)**.
*The involvement of citizens in reducing air pollution positively influences the need to have independent bodies to monitor air pollution.*


**Hypothesis 3 (H3)**.
*Financial incentives for citizens and companies for adopting behaviors that reduce air pollution positively influence the involvement of citizens in reducing air pollution.*


**Hypothesis 4 (H4)**.
*Green investments in the city positively influence the involvement of citizens in reducing air pollution.*


**Hypothesis 5 (H5)**.
*The impact of air pollution on the community directly influences the involvement of authorities in reducing air pollution.*


**Hypothesis 6 (H6)**.
*The impact of air pollution on the community directly influences the involvement of citizens in reducing air pollution.*


**Hypothesis 7 (H7)**.
*The impact of air pollution on the community directly influences the financial incentives for citizens and companies to adopt behaviors that reduce air pollution.*


**Hypothesis 8 (H8)**.
*The impact of air pollution on the community directly influences green investments in the city.*


**Hypothesis 9 (H9)**.
*The impact of air pollution on the community directly influences the need for independent bodies to monitor air pollution.*


We proposed the research model presented in Figure 1, where we included 6 variables and 17 items: the involvement of authorities in reducing air pollution (with four items); the involvement of citizens in reducing air pollution (with two items); financial incentives for citizens and companies to adopt behaviors that reduce air pollution (with two items); green investments in the city (with four items); the air pollution impact on the community (with three items); and the need for independent bodies to monitor air pollution (with two items). The constructs, items, and their codes are detailed in Table 1.

In order to use PLS-SEM [46], we issued a survey to citizens from the most important cities in Romania in September 2022, and received 210 valid questionnaires. The survey included a screening question regarding the environment where the respondents live (urban or rural areas). We focused our research on people living in urban areas because the air quality in rural areas is higher than in big cities, as there are fewer greenhouse gas emissions generated by transportation in rural areas [47,48]. Thus, we eliminated respondents from rural areas, as well as the incomplete surveys, to maintain the consistency of the data. The minimum dimension of the sample when using PLS-SEM is 40 if we consider the 10 times rule, because CTZN is influenced by 4 variables in our model [49]. In the questionnaire, we used the Likert scale (from 1—total disagreement to 5—total agreement) to let respondents express their perception of air quality management. The survey was made with Google Forms and posted online, on the Facebook groups dedicated to people living in the big cities in Romania. It was anonymous and did not collect personal data, thus making people feel safer to express their opinions on these issues. Most respondents are active people between 26 and 45 years old (51.2%), earning more than 800 euros per month (51.5%), who graduated from university (79.1%). In Romania, the net minimum wage is around 300 euros, and the net average wage is around 800 euros [50].

## 4. Results

To check the convergent validity of the items in the proposed model, we calculated the outer loadings and the variance inflation factor (VIF) in Table 2. The items with outer loadings below 0.6 were removed in accordance with studies that show that this level is considered acceptable for convergent validity [51,52]. All VIF (the indicator used for measuring multicollinearity) values for the seventeen items in the model are below four, so the collinearity statistical value is met.

After removing the items with outer loadings below 0.6, the model changes, as shown in Figure 2. The highest impact levels are from POLIMP to MONAIR (0.477), from POLIMP to AUTH (0.461), and from POLIMP to GINV (0.344). FININC, POLIMP, AUTH, and GINV are responsible for 31.7% of the CTZN variance.

We notice from the descriptive statistics in Table 3 that the items we kept in the model have means higher than or equal to 4 (corresponding to agreement and total agreement), exceptPOLIMP3, which has a mean of 3.50.

In Table 4, we calculate Cronbach’s Alpha (the indicator used to measure the consistency of the variables) and average variance extracted (AVE) to check the construct’s reliability and validity. Cronbach’s Alpha is above 0.7 for CTZN, FININC, and MONAIR, which shows high reliability; for AUTH, GINV, and POLIMP, the value is between 0.5 and 0.7, showing moderate reliability [53]. The AVE (the indicator that measures the variance of the constructs) values are higher than 0.5 and the composite reliability is above 0.8, showing the high reliability and validity of the six constructs in the model we proposed.

In Table 5, we applied the Fornell–Larcker criterion and determined the discriminant validity of the research model, because all of the AVE’s square roots for each construct in the model are higher than the other values in the column.

In Table 6, we applied the bootstrapping test to check the model’s significance. Not all path coefficients are significant at a 5% significance level. There are three correlations that do not meet the criteria for t statistics above 1.96 and p-values below 0.05: the impact from CTZN to MONAIR, from GINV to CTZN, and from POLIMP to CTZN. These correlations are also the ones for which the confidence intervals bias corrected include the zero value, which invalidates the hypotheses we developed. The other values meet the requirements for validating the hypotheses. Our conclusions regarding our hypotheses’ validation are included in Table 7. Only hypotheses H1, H3, H5, H7, H8, and H9 are validated.

We used PLSpredict to check the predictive relevance of the research model we proposed. All Q^2^ values in Table 8 are above zero, showing that the model’s constructs have high predictive relevance.

## 5. Discussion

As we mentioned in the previous section, not all of our hypotheses were validated. Therefore, in the following paragraphs, we will discuss each of them.

H1. The involvement of authorities in reducing air pollution positively influences the involvement of citizens in reducing air pollution. This hypothesis was validated, meaning that the legislative measures, controls, and sanctions imposed by public authorities lead to an increase in citizens’ participation in reducing air pollution and changing their behaviors (using public transportation more, and switching to electrical cars). This is in accordance with other studies in the literature [54,55].

H2. The involvement of citizens in reducing air pollution positively influences the need for independent bodies to monitor air pollution. The hypothesis was not validated. Our results did not show a connection between citizen involvement and the need for independent bodies to monitor air pollution. There are no other studies analyzing this direct relationship, but the technical aspects [39,40,41,42,43] required for monitoring sensors might partially explain the results. 

H3. Financial incentives for citizens and companies to adopt behaviors that reduce air pollution positively influence the involvement of citizens in reducing air pollution. This hypothesis was validated, meaning that citizens are more engaged in reducing air pollution if they receive financial incentives. This result is in accordance with other studies [11,13,14,15]. As we can see from Figure 2, the impact of financial incentives on citizen engagement is higher (0.309) than the impact from authorities (0.202).

H4. Green investments in the city positively influence the involvement of citizens in reducing air pollution. This hypothesis was not validated, meaning that citizen engagement is not influenced by green investments made by public and private organizations. Vargas-Hernández et al. [56] appreciate the important role of green spaces in a city, and emphasize the fact that such spaces can influence people to walk more instead of using their cars. Even if this hypothesis was not validated, we can see from Figure 2 that green initiatives have an impact on citizen engagement, even if it is a weak one (0.108).

H5. The impact of air pollution on the community directly influences the involvement of authorities in reducing air pollution. This hypothesis was validated, and there is a strong impact from POLIMP to AUTH (0.461), showing that the negative effects generated by air pollution lead citizens to have an increased perception of authorities’ involvement. The relationship between the effects of pollution on health, quality of life, and some sectors (tourism), and the need for higher involvement of authorities is also presented in other studies [57,58,59].

H6. The impact of air pollution on the community directly influences the involvement of citizens in reducing air pollution. The hypothesis could not be validated and the impact from POLIMP to CTZN is also weak (0.087). This might be explained by citizens’ low level of knowledge regarding the real dangers represented by air pollutants; therefore, more awareness campaigns from authorities might be needed in the future. Rich et al. [60] analyzed the relationship between the health problems generated by pollution and the empowerment effect it might have on some citizens, but highlight the role of public policies focused on the environment and the need for a partnership with the citizens.

H7. The impact of air pollution on the community directly influences the financial incentives for citizens and companies to adopt behaviors that reduce air pollution. This hypothesis was validated, and the impact from POLIMP to FININC is quite high (0.297). The awareness of many problems created by pollution in a community, from reduced quality of life to a decrease in the number of tourists, prompts authorities to offer financial incentives for citizens and companies in order to change their behaviors and shift them to more sustainable ones. This result is consistent with the real burden of pollution for the economy, which leads governments to adopt fiscal policies meant to stimulate green transportation and thus reduce the impact of transport on air pollution. According to the WHO Regional Office for Europe [25], the “annual economic cost of health impacts and mortality from air pollution… stood at USD 1.575 trillion”.

H8. The impact of air pollution on the community directly influences the green investments in the city. The hypothesis was validated, and the impact from POLIMP to GINV is also strong (0.344). Understanding the negative impact of pollution makes public and private organizations invest in creating greener locations and greener buildings, and developing the charging infrastructure for electrical vehicles. Considering the economic cost of pollution [25], the decision to invest in greener technologies to tackle the impact of pollution on the community is both logical and coherent.

H9. The impact of air pollution on the community directly influences the need for independent bodies to monitor air pollution. This hypothesis was also validated and the impact from POLIMP to MONAIR is the strongest in this case (0.477), showing that there is a need for more transparency regarding the level of air pollution in a region, and also that these independent bodies might use different technologies that are more powerful and accurate to measure the levels of air pollution. There are many studies [38,39,40,41,42,43] that prove that sometimes it can be difficult for public authorities alone to install sensors everywhere, considering their limited financial resources and the technologies available for them. The existence of more independent bodies for monitoring air pollution might be a solution for providing data in real time, and might also make people more aware of the health risks of pollution.

## 6. Conclusions

The present study aims to better understand citizens’ perspectives regarding air pollution management. Thus, we analyzed the impact of the following factors: the financial incentives offered by the government; green investments; the health risks posed by air pollution; and the initiatives taken by public authorities to reduce air pollution through legislation, control actions, and sanctions.

The highest impact is from financial incentives that encourage more sustainable forms of transportation, such as the use of electrical cars, and other initiatives taken by authorities that discourage behaviors that produce more air pollution. In total, 75.4% of the respondents in our study mentioned they would use public transportation instead of their personal cars if the transportation were efficient and offered decent travelling conditions. At the same time, 71.5% of them would buy an electrical car if the charging infrastructure were more developed. Moreover, 58.7% of the respondents agree that restricting the circulation of vehicles in the city center in the weekends is a good measure for reducing air pollution. In conclusion, people are willing to use greener forms of transportation if they have the infrastructure and are sufficiently motivated, a result that is consistent with other studies [15].

### 6.1. The Theoretical and Practical Implications

This research is useful for researchers who might extend the study to other countries in order to draw sound conclusions regarding the best strategies that can be used to change behaviors in a community. The results are useful in terms of their practical applicability in the public sector. Managers in local and central authorities can use the results for orienting their strategies and policies to consider the citizens’ perspectives because, if the population is reluctant to adopt the changes, the chances for successful results are low.

### 6.2. Limitations and Future Research Directions

The limits of our study consist in applying the survey online on Facebook groups with members from the most important cities in Romania. The online method for gathering results was preferred because of the limited possibilities of conducting a face-to-face survey in various regions in the country. In future research, we will consider introducing other variables that might influence citizens in their decisions to use greener forms of transportation and contribute to a reduction in air pollution. Thus, an analysis of air pollution management during the COVID-19 pandemic might be considered from both citizens’ perspectives and central and local authorities, as in other studies [61,62]. Another limit that could be tackled in future research relates to the perspective of both public authorities and managers of private companies in Romania. Thus, a broader picture of air pollution management might be offered.

## Figures and Tables

**Figure 1 ijerph-19-14613-f001:**
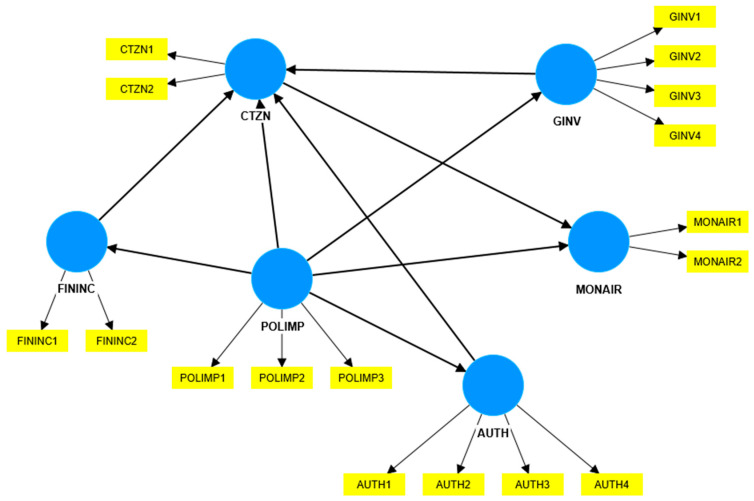
The research model. Source: created with SmartPLS v.4.

**Figure 2 ijerph-19-14613-f002:**
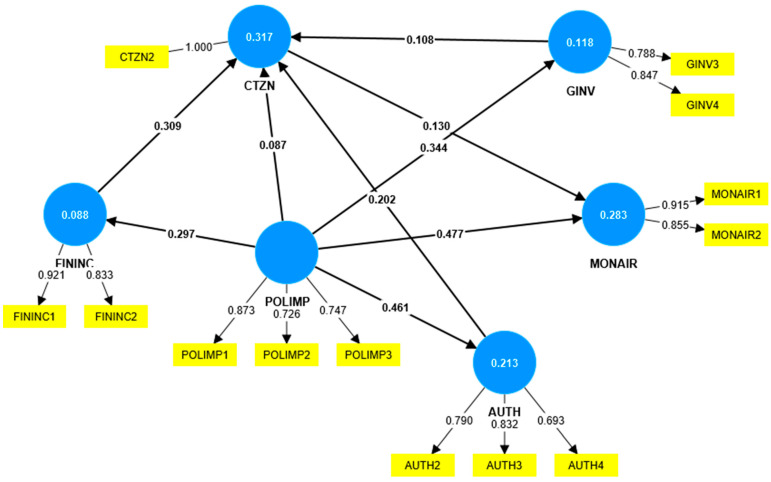
PLS algorithm calculation. Source: created with SmartPLS v.4.

**Table 1 ijerph-19-14613-t001:** The model’s constructs, items, and codes.

Constructs	Items	Codes
Involvement of authorities in reducing air pollution (AUTH)	Authorities are involved in raising the air quality in my city.	AUTH1
	An increase in the control actions implemented by authorities would reduce air pollution.	AUTH2
	The sanctions against those polluting the air should be higher to discourage the phenomenon.	AUTH3
	Restricting the circulation of vehicles on weekends in the city center would reduce air pollution.	AUTH4
Involvement of citizens in reducing air pollution (CTZN)	I actively participate in events related to environmental protection in my community.	CTZN1
	The development of public transportation in terms of efficiency and decent conditions would make me use my car less.	CTZN2
Financial incentives for citizens and companies for adopting behaviors that reduce air pollution (FININC)	Authorities should incentivize citizens and companies to use more eco-friendly vehicles.	FININC1
	Authorities should offer financial incentives to companies that use electrical cars, develop charging stations for them, and switch to green technologies in their activities.	FININC2
Green investments in the city (GINV)	There are sufficient green locations in my city.	GINV1
	The parks in my city contribute to the reduction of air pollution.	GINV2
	New buildings should have horizontal or vertical gardens to reduce air pollution.	GINV3
	Investments in charging stations for electrical vehicles would encourage more citizens and companies to use them.	GINV4
The need for independent bodies to monitor air pollution, in citizens’ opinions (MONAIR)	For more transparency, there is a need for an independent monitorization of air quality.	MONAIR1
	I think an independent network of sensors for air pollution should be developed in cities.	MONAIR2
The air pollution impact on the community (POLIMP)	Air pollution in my city negatively affects the quality of my life.	POLIMP1
	Air pollution in my city leads to an increase in heart and lung problems.	POLIMP2
	Air pollution negatively affects the number of tourists coming to the city.	POLIMP3

**Table 2 ijerph-19-14613-t002:** Outer loadings and VIF values.

Items	Outer Loadings	VIF
AUTH1	−0.328	1.070
AUTH2	0.791	1.558
AUTH3	0.836	1.626
AUTH4	0.673	1.131
CTZN1	0.458	1.013
CTZN2	0.934	1.013
FININC1	0.916	1.438
FININC2	0.840	1.438
GINV1	−0.110	1.035
GINV2	0.344	1.029
GINV3	0.758	1.135
GINV4	0.797	1.135
MONAIR1	0.912	1.490
MONAIR2	0.859	1.490
POLIMP1	0.873	1.599
POLIMP2	0.726	1.348
POLIMP3	0.748	1.296

Source: Authors’ analysis using SmartPLS v.4.

**Table 3 ijerph-19-14613-t003:** Descriptive statistics for the items in the research model.

Items	Mean	Standard Deviation	Outer Loading
AUTH2	4.47	0.91	0.790
AUTH3	4.67	0.81	0.832
AUTH4	4.00	1.38	0.693
CTZN2	4.25	1.17	1.000
FININC1	4.52	0.91	0.921
FININC2	4.30	1.12	0.833
GINV3	4.25	1.05	0.788
GINV4	4.08	1.22	0.847
MONAIR1	4.61	0.87	0.915
MONAIR2	4.46	1.00	0.855
POLIMP1	4.14	1.13	0.873
POLIMP2	4.62	0.85	0.726
POLIMP3	3.50	1.40	0.747

Source: Calculated with SmartPLS v.4.

**Table 4 ijerph-19-14613-t004:** Construct reliability and validity.

Construct	Cronbach’s Alpha	rho_A	Composite Reliability	AVE
AUTH	0.661	0.662	0.817	0.599
CTZN	1	1	1	1
FININC	0.711	0.771	0.871	0.771
GINV	0.507	0.514	0.801	0.669
MONAIR	0.729	0.760	0.879	0.784
POLIMP	0.687	0.713	0.827	0.616

Source: Calculated with SmartPLS v.4.

**Table 5 ijerph-19-14613-t005:** Fornell–Larcker criterion.

Construct	AUTH	CTZN	FININC	GINV	MONAIR	POLIMP
AUTH	0.774					
CTZN	0.444	1.000				
FININC	0.495	0.501	0.878			
GINV	0.450	0.418	0.614	0.818		
MONAIR	0.566	0.278	0.378	0.355	0.886	
POLIMP	0.461	0.309	0.297	0.344	0.517	0.785

Source: Calculated with SmartPLS v.4.

**Table 6 ijerph-19-14613-t006:** Bootstrapping test.

	T Statistics	*p*-Values	Confidence Interval Bias Corrected
AUTH −> CTZN	2.186	0.029	(0.023, 0.381)
CTZN −> MONAIR	1.645	0.100	(−0.023, 0.286)
FININC −> CTZN	2.803	0.005	(0.107, 0.537)
GINV −> CTZN	1.117	0.264	(−0.081, 0.292)
POLIMP −> AUTH	6.675	0.000	(0.311, 0.585)
POLIMP −> CTZN	0.953	0.341	(−0.088, 0.264)
POLIMP −> FININC	3.930	0.000	(0.136, 0.433)
POLIMP −> GINV	5.164	0.000	(0.195, 0.461)
POLIMP −> MONAIR	5.745	0.000	(0.295, 0.622)

Source: Calculated with SmartPLS v.4.

**Table 7 ijerph-19-14613-t007:** Hypotheses’ validation.

Hypothesis	Validation
AUTH −> CTZN (H1)	Validated
CTZN −> MONAIR (H2)	Invalidated
FININC −> CTZN (H3)	Validated
GINV −> CTZN (H4)	Invalidated
POLIMP −> AUTH (H5)	Validated
POLIMP −> CTZN (H6)	Invalidated
POLIMP −> FININC (H7)	Validated
POLIMP −> GINV (H8)	Validated
POLIMP −> MONAIR (H9)	Validated

Source: Authors’ own analysis.

**Table 8 ijerph-19-14613-t008:** PLSpredict results.

Construct	Q^2^ values
AUTH	0.192
CTZN	0.077
FININC	0.071
GINV	0.103
MONAIR	0.246
POLIMP	-

Source: Calculated with SmartPLS v.4.

## Data Availability

Not applicable.

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
