# Peer review of "Air Pollution Management: A Multivariate Analysis of Citizens’ Perspectives and Their Willingness to Use Greener Forms of Transportation"

_ijerph, 2022, doi:10.3390/ijerph192114613_

Round 1
Reviewer 1 Report
The authors aim to understand the way air pollution can be managed by public authorities. The manuscript is generally good, contains complete content, the quality of figures and tables is satisfactory. The study has social value in providing the role played by the citizens’ perspective for establishing strategies that will be easier to implement and for which the reluctance to change will be reduced. Some suggestions on the author's research are given here.
Main problems:
1. The abstract of this article does not explain the specific research content clearly.
2. In the actual use of the questionnaire survey method, there are often problems such as miscellaneous data, difficult data recovery, insufficient data and low data quality. The article should point out how to screen data to select effective questionnaires.
3. The role of the proposed framework in the management of air pollution effect was not well described in the article overview. Here, examples are needed.
4. L55-62: Whether these variables broadly include the perspective of citizens, and whether there are omissions (such as government punitive measures, etc.).
5. L93-95: What does the authors consider that financial incentives and the support offered to the citizens and companies are having a more powerful impact than punitive measures on the citizens’ change in behavior based on? Does the authors’ view contradict the previous related research that public managers should adopt more strict regulations to reduce air pollution and raise air quality?
Minor problems:
1. (Figure 2) The circle corresponding to POLIMP is not marked with corresponding numbers.
2. (Table 8) The descriptive statistical analysis results of each item will be appropriately placed on after Table 1.
3. L128-129: The authors chose to survey on citizens from the most important cities in Romania, but in general, the more important cities are, the better they are at independent monitoring of air quality. Did the authors consider that this might affect the MONAIR part of the survey results?
4. L147-149: Is there any literature on studies that show pollution increases costs to health systems?
5. L176-177: What exactly is the solution?
6. Table1: What is the difference between MOMAIR 1 and MONAIR 2?
7. It is suggested to add some references in recent years, at the same time, quote the impact of government measures on pollutants during the COVID-19 epidemic (such as: https://doi.org/10.3390/atmos12020205, doi:10.3389/fenvs.2021.806094), and increase the balance between the government's pollutant control measures and social and economic benefits in the discussion section.
Author Response
Dear Reviewer,
Thank you for your observations and for the opportunity to improve our manuscript!
We are very grateful for taking the time to analyze the paper and make very useful, encouraging and thoughtful comments and recommendations.
We have read the evaluation carefully and, based on the review reports, we performed significant revisions of our manuscript, as requested, highlighted with red into the manuscript, respectively:
Main problems:
- The abstract of this article does not explain the specific research content clearly.
Response: We added a phrase in the abstract to better highlight the variables that were used in order to understand citizens perspective and thus being able to formulate theoretical and practical implications for public administration. They should have been mentioned in the abstract indeed.
- In the actual use of the questionnaire survey method, there are often problems such as miscellaneous data, difficult data recovery, insufficient data and low data quality. The article should point out how to screen data to select effective questionnaires.
Response:
We added a paragraph in the Method section to explain the valid questionnaires we kept for the research. A screening question regarding the areas where the respondents live was used and we also checked for incomplete questionnaires.
More references were added here. We also explained the dimension of the survey in accordance with the method we used.
- The role of the proposed framework in the management of air pollution effect was not well described in the article overview. Here, examples are needed.
Response:
A paragraph was added to better emphasize the role of the framework we proposed in the research and also highlighted examples from Bangladesh, China and Antwerp which were analyzed by other researchers showing the role played by the citizen:
“The novelty of our research consists in our focus on citizens’ perspective regarding air pollution management. We believe that this perspective can help authorities in raising the level of compliance when establishing public policies meant to protect the environment. Even if the proposed perspective is not analyzed by other researchers in its entirety, many studies present the success of environmental policies when authorities take into account citizens. Various countries might require different actions: Mahmood [9] shows that more strict regulations are needed in Bangladesh to make citizens comply; Wang and Wheeler [11] studied the connection between fiscal incentives and the changes in the citizens’ behavior and the effect on air quality in China noticing the direct and positive relationship; Van Brussel and Huyse [16] also studied the role played by citizens in reducing air pollution in Antwerp. All these examples show the important role played by the citizen and the fact that authorities should construct their policies and strategies starting from the population”.
- L55-62: Whether these variables broadly include the perspective of citizens, and whether there are omissions (such as government punitive measures, etc.).
Response: We added the punitive measures within the explanation provided for the first variable which reflects the passive involvement. The specific explanations for the variables are provided in sections 2.1-2.6.
- L93-95: What does the authors consider that financial incentives and the support offered to the citizens and companies are having a more powerful impact than punitive measures on the citizens’ change in behavior based on? Does the authors’ view contradict the previous related research that public managers should adopt more strict regulations to reduce air pollution and raise air quality?
Response: Thank you for your observation. We changed the word consider with assume because this assumption determined us to formulate hypotheses H1 and H3. Both punitive and incentives are more important but our research revealed that incentives are having a higher impact on citizens than punitive measures. The explanation is in the Results and Discussion section. We did not find comparative studies between them in the literature but we started with this assumption. And probably consider was not a good choice to express that. So, thank you for noticing the nuance.
Minor problems:
- (Figure 2) The circle corresponding to POLIMP is not marked with corresponding numbers.
Response: POLIMP is an independent variable in our model so there is no number in the circle. The figure is provided based on Smart-PLS application.
- (Table 8) The descriptive statistical analysis results of each item will be appropriately placed on after Table 1.
Response: We moved it after table 2 because the model changed after eliminating some items with the outer loadings below 0.6 in table 2.
- L128-129: The authors chose to survey on citizens from the most important cities in Romania, but in general, the more important cities are, the better they are at independent monitoring of air quality. Did the authors consider that this might affect the MONAIR part of the survey results?
Response: MONAIR refers to the perception of citizens regarding the need for monitoring air quality. The perception does not take into account the real presence of sensors or independent bodies.
- L147-149: Is there any literature on studies that show pollution increases costs to health systems?
Response: Yes. The reference was in conclusion but we moved it in the literature review. The reference is: WHO Regional Office for Europe. Economic cost of the health impact of air pollution in Europe: Clean air, health and wealth. WHO Regional Office for Europe: Copenhagen, Denmark, 2015. Available online: https://www.euro.who.int/__data/assets/pdf_file/0004/276772/Economic-cost-health-impact-air-pollution-en.pdf (accessed on 16 October 2022)
- L176-177: What exactly is the solution?
Response: We added the solution there, which is represented by “unmanned aerial vehicles”.
- Table1: What is the difference between MOMAIR 1 and MONAIR 2?
Response: In table 1, MONAIR1 refers to the importance of monitoring to ensure transparency because there are also some public owned sensors but the population might not trust them. So, the question for MONAIR1 puts the focus on the need for transparency and MONAIR2 asks citizens about the need for these monitoring sensors in the first place. For example, I might agree that there should be independent bodies to monitor air quality because I do not trust authorities but maybe the importance I give to these sensors/independent bodies is not so high compared with the importance for fiscal incentives for example. And this connection was intended here. And Hypothesis 2 was not supported in our research.
- It is suggested to add some references in recent years, at the same time, quote the impact of government measures on pollutants during the COVID-19 epidemic (such as: https://doi.org/10.3390/atmos12020205, doi:10.3389/fenvs.2021.806094), and increase the balance between the government's pollutant control measures and social and economic benefits in the discussion section.
Response: The following references were added in the conclusion section in the paragraph regarding future research directions. In the Discussion we kept only studies regarding the variables we considered and discussed if other studies had similar results or not. Also, references number 6,10,14, 21, 36, 50 and 58 are from 2021-2022 and could be seen in the Discussion section too.
Dong, L.; Chen, B.; Huang, Y.; Song, Z.; Yang, T. Analysis on the Characteristics of Air Pollution in China during the COVID-19 Outbreak. Atmosphere 2021, 12, 205. https://doi.org/10.3390/atmos12020205
Chen B, Huang Y, Huang J, Dong L, Guan X, Ge J, Hu Z. Using Lidar and Historical Similar Meteorological Fields to Evaluate the Impact of Anthropogenic Control on Dust Weather During COVID-19. Frontiers in Environmental Science, 2021, 619. https://doi.org/10.3389/fenvs.2021.806094

Reviewer 2 Report
This work, aimed at understanding the way air pollution can be managedby public authorities, both central and local, starting from the citizens’ perspective
on the issue, confirms other studies previously performed in other regions. After reading the paper carefully, I have some suggestions. First, I would suggest adding the PLS-SEM model description (as well as a reference)
and the description of the analysis method implemented in order to simplify
the understanding for less experienced readers.
In this regard, I would suggest adding a specific section for the description
of the method and analysis.
I would have found it useful:
line 230: to define “VIF” and “outer loading”
line 244: to define “AVE” and “Cronbach’s Alpha”
line 248: to define “Composite Reliability”
line 252: to define “Fornell-Larcker criterion”
line 270: to define “Q^2”.
As regards the references, there isn’t correspondence with the bibliography.
I would also suggest combining Tables 5 and 6 in a single table.
Some possible oversight errors:
line 70: pasive à could be “passive”
line 306: week à could be “weak”
One last thing: As for the sample size, are you sure that the data used (210 - line 219)are sufficient to describe the entire region? How many big cities are considered? If the considered big cities in Romania are 5,
then the dataset consists of 40 questionnaires per city.
It is safe to assume that the analyzed sample isn’t representative
of the described population.
Author Response
Dear Reviewer,
Thank you for your observations and for the opportunity to improve our manuscript!
We are very grateful for taking the time to analyze the paper and make very useful, encouraging and thoughtful comments and recommendations.
We have read the evaluation carefully and, based on the review reports, we performed significant revisions of our manuscript, as requested, highlighted with red into the manuscript, respectively:
This work, aimed at understanding the way air pollution can be managed
by public authorities, both central and local, starting from the citizens’ perspective
on the issue, confirms other studies previously performed in other regions. After reading the paper carefully, I have some suggestions.
- First, I would suggest adding the PLS-SEM model description(as well as a reference)
and the description of the analysis method implemented in order to simplify
the understanding for less experienced readers.
In this regard, I would suggest adding a specific section for the description
of the method and analysis.
I would have found it useful:
line 230: to define “VIF” and “outer loading”, line 244: to define “AVE” and “Cronbach’s Alpha”
line 248: to define “Composite Reliability”, line 252: to define “Fornell-Larcker criterion”, line 270: to define “Q^2”.
Response:
We included a few more references in the Method section to explain the method we used. Also, we moved table 8 after table 2 which we think make things clearer for the readers.
- Partial Least Squares Structural Equation Modeling: An Emerging Tool in Research. Available online: https://www.methodspace.com/blog/partial-least-squares-structural-equation-modeling-emerging-tool-research (accessed on 01 November 2022)
- Kock N., Hadaya P. Minimum sample size estimation in PLS‐SEM: The inverse square root and gamma‐exponential methods. Information systems journal 2018, 28(1), pp. 227-61. https://doi.org/10.1111/isj.12131.
Values in tables 2-6 and 8 were calculated with Smart-PLS software which was mentioned in the research methodology section. All indicators in these tables are calculated with the software and the values were briefly explained to show the statistical relevance of the model we used.
More explanations for the indicators would not bring more value to the paper and would create more confusion to the usual reader because they are very specific to statistics, mathematics and not related to the topic of our research. Also, they are so general and in the common language of statistics that it will raise the similarity ratio of our paper.
Thus, VIF is variance inflation factor, measuring the collinearity of the regression. Outer loadings are the estimated relationships in reflective models. Average variance extracted (AVE) – measure the variance of a construct. Cronbach’s Alpha measurers the internal consistency as well as composite reliability in scale items like those used by us in the model. Fornell-Larcker criterion measures the variance shared by the variables in the model. And Q2 measures the predictive relevance of the model.
PLS method has specific steps to follow and it is difficult to add an extensive explanation of the method and all of the indicators in our paper.
Still, we included a few references that explain the method per se and the relevance of a value or another for the indicators we calculated with the Smart-PLS software.
These are useful for the readers wanting to know more about the method and they are included in the text and in the reference list:
- Ringle, C.M., Wende, S. and Becker, J.M. (2015). SmartPLS 4. Boenningstedt: SmartPLS GmbH. Available online: http://www.smartpls.com (accessed on 11 October 2022).
- Hulland, J. Use of partial least squares (PLS) in strategic management research: A review of four recent studies. Strategic management journal 1999, 20(2), pp. 195-204.
- Hinton, P., McMurray, I., and Brownlow, C. SPSS Explained, 2nd ed., Routledge: London, UK, 2014. https://doi.org/10.4324/9781315797298
- Yana, A.G.A., Rusdhi, H.A. and Wibowo, M.A. Analysis of factors affecting design changes in construction project with Partial Least Square (PLS). Procedia Engineering 2015, 125, pp. 40-45.
- As regards the references, there isn’t correspondence with the bibliography.
Response: We added more references and checked the correspondence and the number of the references.
- I would also suggest combining Tables 5 and 6 in a single table.
Response: Thank you for the suggestion. I understand your point of view, but we think that it would be better to leave them separate because table 5 presents the results provided by the software Smart-PLS and for keeping the steps of the applied method consistent, we do not want to include our analysis in the table providing the Bootstrapping test.
- Some possible oversight errors: line 70: pasive à could be “passive”, line 306: week à could be “weak”
Response: We really missed those. We appreciate the keen eye. Thank you.
- As for the sample size, are you sure that the data used (210 - line 219)
are sufficient to describe the entire region? How many big cities are considered? If the considered big cities in Romania are 5,
then the dataset consists of 40 questionnaires per city.
It is safe to assume that the analyzed sample isn’t representative
of the described population.
Response: There was added a reference in the method section to explain the minimum number required by PLS-SEM method which is 10 times the number of links pointing to a construct. In our case, CTZN is the construct with the highest number of links (4). Even if the minimum number required is 40 and in fact, this is the reason the method is preferred because of its simplicity and the fact that is used for small samples, we decided to have a higher number of responses. After eliminating the incomplete responses and people in rural areas, we remained with 210 valid answers.
The reference is: Kock N., Hadaya P. Minimum sample size estimation in PLS‐SEM: The inverse square root and gamma‐exponential methods. Information systems journal 2018, 28(1), pp. 227-61. https://doi.org/10.1111/isj.12131.

Round 2
Reviewer 1 Report
Minor language and text editing errors.
Author Response
Dear Reviewer,
Thank you for your observations and for the opportunity to improve our manuscript!
We are very grateful for taking the time to analyze the paper and make very useful, encouraging and thoughtful comments and recommendations.
We have read the evaluation and, based on the review reports, we performed revisions of our manuscript, as requested, highlighted with red into the manuscript.
- Minor language and text editing errors.
Response. The text was checked again and all errors were corrected.

Reviewer 2 Report
In this analysis, parameters are estimated (such as VIF, Cronbach's Alpha, and AVE) but no description is given. In this way the work is not self-consistent. The reader must look elsewhere for the information necessary to interpret the results.
Author Response
Dear Reviewer,
Thank you for your observation and for the opportunity to improve our manuscript!
We are very grateful for taking the time to analyze the paper and make very useful, encouraging and thoughtful comments and recommendations.
We have read the evaluation and, based on the review reports, we performed revisions of our manuscript, as requested, highlighted with red into the manuscript, respectively:
- In this analysis, parameters are estimated (such as VIF, Cronbach's Alpha, and AVE) but no description is given.
Response:
We included a description of the indicators you mentioned. The description for VIF, AVE and Cronbach's Alpha is highlighted in red, on lines 259-260, 278-279, 282-283.
